# Framework for Evaluating Potential Causes of Health Risk Factors Using Average Treatment Effect and Uplift Modelling

**Daniela Galatro** [1], **Rosario Trigo-Ferre** [2], **Allana Nakashook-Zettler** [1], **Vincenzo Costanzo-Alvarez** [3,*], **Melanie Jeffrey** [4], **Maria Jacome** [5], **Jason Bazylak** [3] and **Cristina H. Amon** [1,3]

1. Department of Chemical Engineering and Applied Chemistry, University of Toronto, Toronto, ON M5S 3E5, Canada
2. Faculty of Applied Science and Engineering, University of Toronto, Toronto, ON M5S 3E5, Canada
3. Department of Mechanical and Industrial Engineering, University of Toronto, Toronto, ON M5S 3G8, Canada
4. Centre for Indigenous Studies, University of Toronto, Toronto, ON M5S 2J7, Canada
5. Faculty of Applied Sciences and Technology, Humber Institute of Technology and Advanced Learning, Toronto, ON M9W 5L7, Canada
* Correspondence: v.costanzo@utoronto.ca

**Abstract:** Acute myeloid leukemia (AML) is a type of blood cancer that affects both adults and children. Benzene exposure has been reported to increase the risk of developing AML in children. The assessment of the potential relationship between environmental benzene exposure and childhood has been documented in the literature using odds ratios and/or risk ratios, with data fitted to unconditional logistic regression. A common feature of the studies involving relationships between environmental risk factors and health outcomes is the lack of proper analysis to evidence causation. Although statistical causal analysis is commonly used to determine causation by evaluating a distribution's parameters, it is challenging to infer causation in complex systems from single correlation coefficients. Machine learning (ML) approaches, based on causal pattern recognition, can provide an accurate alternative to model counterfactual scenarios. In this work, we propose a framework using average treatment effect (ATE) and Uplift modeling to evidence causation when relating exposure to benzene indoors and outdoors to childhood AML, effectively predicting causation when exposed indoors to this contaminant. An analysis of the assumptions, cross-validation, sample size, and interaction between predictors are also provided, guiding future works looking at the universalization of this approach in predicting health outcomes.

**Keywords:** acute myeloid leukemia; risk factors; average treatment effect; uplift modelling; machine learning; benzene

## 1. Introduction

Acute myeloid leukemia (AML) is a cancer of the myeloid line of blood cells; AML starts in the blood stem cells and is characterized by its rapid growth [1]. While AML is the most common type of leukemia in adults, it also affects children; and about 500 children are diagnosed with AML in the U.S. annually [2]. Childhood AML is most prevalent during the first two years of life and adolescence. Epidemiological and genetic studies have confirmed that most infant leukemias develop in utero [3,4].

Chemical exposure to significant benzene concentrations is reported as a possible cause of AML in occupationally exposed workers [5]. Benzene exposure has also been reported to increase the risk of developing AML in children. Most existing reports are retrospective case-control studies [6], which are inherently limited since benzene exposure is typically measured indirectly (biased) as parents of sick versus healthy children may differentially recall them [5].

Moreover, some positive findings may be due to confounding factors instead, as other biases are added when specific segments of the population are under-represented in the study or control cohorts [6]. It has also been noted that exposure to various solvents

and hydrocarbons increases the chance of developing childhood AML [7,8]. Both groups of chemicals fall into a vast range of toxicological profiles, benzene being the one of greatest concern.

In the following subsections, we briefly review the traditional approaches to establishing the relationship between Benzene Exposure and AML and current trends in estimating causation for health outcomes using Machine Learning outcomes.

### 1.1. Traditional Approaches to Establishing the Relationship between Benzene Exposure and AML

The Bradford Hill criteria [9,10] have been extensively used to evaluate causation when human epidemiologic relationships are found between exposure to a contaminant such as benzene and the disease outcome, such as AML or other hematopoietic and pulmonary diseases [10]. Despite its extensive use, this method is still debated among epidemiologists, as they question, among many arguments, its scope of application and the possibility of ruling out causality in some specific scenarios.

The assessment of the potential relationship between environmental benzene exposure and childhood AML includes studying exposures prior to conception, during pregnancy, or while breastfeeding [5]. Parental exposure to benzene, for instance, has been studied as a potential risk factor for infant AML, with conflicting results among researchers. Kaatsch et al. [11] and Shu et al. [12] did not find any association between parental exposure and childhood AML. On the other hand, Buckley et al. [13] and Magnani et al. [14] reported elevated rates of childhood AML associated with benzene, solvents, and paternal petroleum occupational exposure (prenatal).

Households' exposures to benzene have also been investigated, showing no increases in childhood AML related to home use of solvents [15,16]. Nevertheless, cigarette smoke, the main non-occupational source of benzene, is associated with an increased risk of developing this disease, even at low-level exposure to benzene, through parental smoking during pregnancy [17,18].

Other exposure surrogates for benzene as an air pollutant are traffic density and proximity to chemical plants, refineries, and gas stations. While several air pollutants are typically present in different concentrations, their risks are ranked by odds ratios. Norlinder and Jarkvolm [19] observed an increase in childhood AML related to car density with an incidence greater than 20 cars/km$^2$, possibly attributable to benzene in the gasoline. Reynolds [20] reported correlations between traffic density between benzene and butadiene air concentrations. The corresponding odds ratios for children reveal a relationship with the incidence rate of childhood AML. Similarly, Crosignani et al. [21] also reported an association between elevated rates of childhood AML to traffic density, attributable to high benzene exposure. Steffan et al. [22] reported an increased risk of this disease in proximity to gas stations, while Harrison et al. [23] found no association. These opposing results have also been reported when analyzing the risks of nearby petrochemical plants [5].

The discrepancies found in the literature may be due to the method of estimating exposures, potential confound from correlated pollutants, and the intrinsically anisotropic nature of the control volumes defined for the studies. For instance, air measurements in outdoor volumes, show considerable variability in space and time, compared to values reported for indoor control volumes. Benzene has undoubtedly been classified as a human carcinogen causing AML in adults when exposed to relatively high concentrations at work. Nevertheless, it is not clear if low concentrations of this compound in outdoor air cause childhood AML, although some studies reveal some association [24].

Previous studies involving benzene exposure and childhood AML statistically reported their results as odds ratio (OR) and/or risk ratio (RR). OR stands for the ratio of an event's odds in one group (for example, the exposed group) to its odds in the other group (for example, the nonexposed group); the risk ratio (RR) stands for the ratio of the risk of an event in one group to the risk of an event in the other group. An OR or RR greater than 1.0 indicates an increase in odds or risk among the exposed group compared to the unexposed one. Conditional and unconditional logistic regressions are typically

used to fit the data of these studies, hence estimating the ORs, and RRs [25–27]. Matching is used in case-control studies to adjust for confounding data, ensuring that regression is possible when there is not enough overlap in confounding variables between cases and a set of controls. Age, sex, and race are typical confounders suggested by descriptive epidemiology [28]. Since the distribution of these variables may differ between cases and controls, matching is commonly used to select cases and controls with similar distributions of similar variables. Unmatched case-control studies, on the other hand, are typically analyzed using unconditional logistic regression (ULR) due to their robust estimates and effectiveness if there are few confounders to adjust for [23,27]. ULR involves producing exposure-disease strata for each level of the confounder and then producing an average effect across the data. This method is also effective when there are no problems with sparse data, no loss of validity, and a potential increase in precision [29].

ULR makes the events log-odds a linear mixture of one or more independent factors, predicting the likelihood that an event will occur [30]. Similar to other regressions, this method shows where there is a relationship between these variables. Nevertheless, this relationship cannot always imply causation; in other words, regression does not determine causation. In statistics, the causal analysis goes one step further than the standard statistical analysis in assessing the parameters of a distribution, aiming to infer probabilities under changing conditions and understanding the actual effect of a specific phenomenon happening in a system.

*1.2. Current Trends in Estimating Causation for Health Outcomes Using Machine Learning Methods*

Causal analysis can be sometimes inaccurate, as in complex systems, it is difficult to make causal arguments based on single correlation coefficients. To overcome this challenge, recent machine learning (ML) approaches have been developed, based on causal pattern recognition, providing robustness in accurately modeling counterfactual scenarios [31].

ML methods have been applied to health sciences for causal inference [32]. For counterfactual prediction, ML has been used to address causal questions using methods such as Random Forest [33] and Bayesian additive regression trees (BART) [34]. Some common research goals that can be tackled with these techniques are [30] (i) the evaluation of potential causes of health outcomes, (ii) the assessment of treatment options, and (iii) the assessment of bias in the statistical analysis. Some ML predictors proved to be assertive when causally inferring the influence of a health outcome while being controlled by confounders. One study, for instance, showed that the targeted maximum likelihood estimation (TMLE) technique outperformed traditional models when analyzing the effect of fruit density on the nutrition of pregnant women on birth outcomes [35]. Despite being a promising alternative to causal analysis, it is still in the foreseeable future when we will be able to identify high-level causal variables from low-level data using these techniques, as causal modeling approaches such as meta-learning and meta-modeling will be able to find causal relationships accurately. Moreover, the potential case of being exposed to complex mixtures of chemical contaminants causing adverse health outcomes can have additive or synergistic effects, posing a challenge where the strength of the ML approach could be used in combination with existing human data to infer causality [36].

Uplift modeling has been used by several companies to estimate the effect of an action on some customer outcomes [37]. Estimating customer uplift is a causal inference since it requires determining the difference between two outcomes that are mutually exclusive for an individual (counterfactual nature); this is carried out using randomized experiments for the treatment group and the control group. At the same time, the Uplift estimation is also a machine learning problem because different models must be trained to finally select the one that yields the most reliable prediction, requiring a sensible cross-validation process [38]. The combination of these features taken from both approaches, causal inference, and machine learning, make Uplift modeling a suitable alternative in determining causation. To our knowledge, Uplift modeling has not been applied to health sciences. Uplift modeling

might become unstable when predicting causation, as with any other ML technique, as the sample size decreases. Moreover, its convergence might be affected since it can depend on variables not typically used in response models.

In this work, we propose a novel framework, for evaluating potential causes of health outcomes using the average treatment effect (ATE) to compare the effects in our computational experiments and Uplift modeling, as a machine learning technique employed for cross-validation of the ATEs. This simple approach effectively estimates the causation of health outcomes using two levels of confirmation, integrating causal inference and machine learning features. Our case study includes the causation of benzene exposure (as an air pollutant) to childhood AML, analyzing the counterfactual nature of the OR-based relationships found indoors and outdoors. The case study selected for our framework was presented by Heck et al. [25] and includes childhood AML data collected from California birth records over 17 years. Health outcomes based solely on relationships between variables can be challenged when the outcome is not proven to result from the occurrence of the other event (s). This paper aims to describe the development and use of this framework and discuss its significance as a reliable causation framework that can potentially be effectively used to prevent, monitor, and treat diseases.

## 2. Methods

The methodological workflow (Figure 1) can be applied as a framework for any study involving the causation of a pollutant exposure to the risk of developing a disease through machine learning techniques and the analysis of the counterfactual nature of the OR-based relationships.

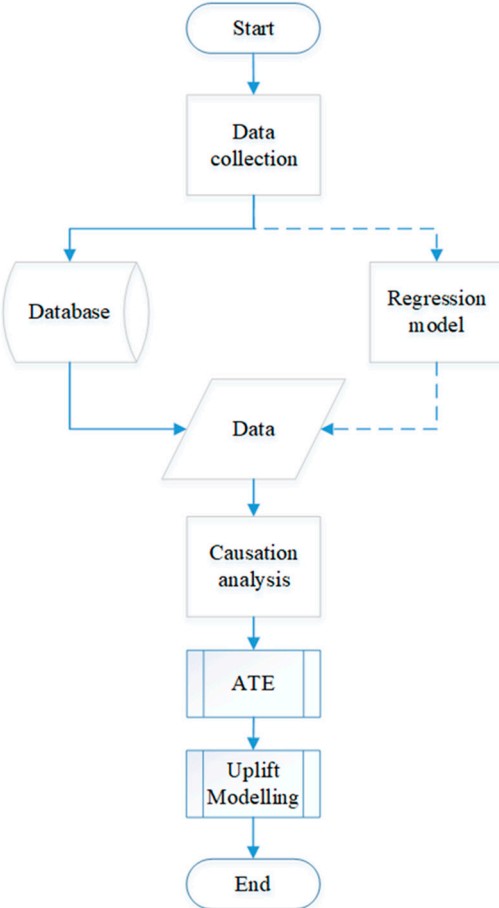

**Figure 1.** Framework for causation analysis of a pollutant exposure to the risk of developing a disease using machine learning techniques.

The proposed ATE-Uplift framework is depicted in Figure 1, where the data is collected either from an existing database or by obtaining simulated data from a regression model. Typically, health outcome studies in the literature comprise strata of data that are fitted using logistic regressions, as discussed in Section 2.1. Once collected, our causation analysis includes two stages: (i) estimating the ATEs and (ii) cross-validating the previous estimation using the Uplift modeling. The two levels strategy based on cross-validation integrating causal inference and machine learning is crucial to support the reliability of our framework since the consequences of determining causation can have several implications in the prevention, monitoring, and treatment of diseases. Ultimately, our framework provides two integrated causation indicators, the ATE values, and Qini coefficients. The results from these stages allow researchers to conclude the causation of pollutant exposure to the risk of developing a disease.

### 2.1. Case Study

The data for the case study selected in this work was presented by Heck et al. [25]. In their work, they ascertained 46 cases of AML from the California Cancer Registry records of children with ages less than six years and 19,209 controls from California birth records between 1990 and 2007 and within 6 km of air monitoring stations. Risks of developing AML were reported as ORs, including the ones associated with exposure to benzene, butadiene, toluene, and other air pollutants, concluding that there is an increased risk for AML when exposed to these air pollutants. A 1-M-matched case-control study was performed using unconditional logistic regression (ULR), adjusted for the year of birth as a matching variable, with 1:M = 1:20, to compare several characteristics of cases and controls, primarily demographics (socioeconomic status, race, birthplace, and parity). Then, ULR was employed to estimate the risk of AML from each interquartile range increase in air toxic exposure for the different pollutants separately. The data might require evaluating the correlations between pollutants. Heck et al. [25] recommended performing a factor analysis with varimax rotation to group highly correlated pollutants based on eigenvalues greater than 1. We have selected the OR related to exposure to benzene, which reveals a relationship between this exposure and the risk of developing childhood AML.

### 2.2. Data Generation and Preparation for Uplift

This section summarizes the process used to generate data from the 1-M matched control case study fitted with URL [23], as the original numerical and categorical data of the study was unavailable, and its consequent preparation for ATE-Uplift, including factual and counterfactuals. The data generation involves drawing samples from the existing ULR model by Heck et al. [23].

Let $y$ be the case-control study, where $y = 1$ for a case and $y = 0$ for the control. Let $x_m = \{x_{m1}, x_{m2}\}$ a vector of matching variables, $x_e$ and exposure associated with the case-control status, and $x_0$ a vector of unmatched dummy variables corresponding to the strata of the matched variables. The ULR model, assuming no interaction between the predictors, is given by,

$$logit(\pi) = \beta_0 + \beta_e x_e + \beta_m^T x_m + \beta_0^T x_0 \tag{1}$$

where $\pi$ is the probability of developing the disease, and $\beta$'s are the regression coefficients. $T$ denotes transpose.

The input data for the ATE model must be generated as a table of factual and counterfactual cases. 'Factuals' are persons in the factual universe (where a 'zero' value is assigned to benzene, or benzene = 0), and 'counterfactuals' are persons in a counterfactual universe (where 'one' value is assigned to benzene, or benzene = 1). The table is generated from the expected propensity of AML conditional on the inputs, which are predicted and converted into a binary variable, as shown in Table 1.

**Table 1.** Factuals and counterfactuals input for the ATE model.

| Person | Socioeconomic Status | | | | Race | | Birthplace | Parity | Benzene | BINaml |
|---|---|---|---|---|---|---|---|---|---|---|
| 1 | 0 | 0 | 0 | 0 | 0 | 0 | 0 | 0 | 0 | 0 |
| 2 | 0 | 0 | 0 | 0 | 0 | 0 | 0 | 1 | 0 | 0 |
| 3 | 0 | 0 | 0 | 0 | 0 | 0 | 1 | 0 | 0 | 0 |
| ... | ... | ... | ... | ... | ... | ... | ... | ... | ... | ... |
| 61 | 0 | 0 | 0 | 0 | 0 | 0 | 0 | 0 | 1 | 0 |
| 62 | 0 | 0 | 0 | 0 | 0 | 0 | 0 | 1 | 1 | 0 |
| 63 | 0 | 0 | 0 | 0 | 0 | 0 | 1 | 0 | 1 | 0 |

In Table 1, person 1 and person 61 are identically theoretical persons, except for having different binary exposures to benzene and potentially different binary AML outcomes. Similarly, and with the same proviso, person 2 and person 62 are identical, person 3 and person 63 are identical, and so on.

We used simulated data from the ULR of the case study; the simulation assumes no interaction between the benzene exposure and the rest of the predictors (socioeconomic status, race, birthplace, and parity); hence, Simpson's paradox is ruled out. This paradox emerges when groups of data show one trend, but this is reversed when the groups are combined. Our factual and counterfactuals table, in particular, the BINaml column (the presence or absence of AML), is generated by the projection of the target variable of Equation (1) or its substitute, Equation (2), that is $logit(\pi(x))$ of AML and its associated binary AML variable.

$$logit(\pi(x)) = \beta_0 + \beta_e x_e + \beta_m^T x_m \qquad (2)$$

This projection requires Equation (1) to be completely specified, including the intercepts, which was omitted by Heck et al. [25] as is usual in matched case-controlled studies. Therefore, we substitute Equation (1) with Equation (2) to estimate the intercept, where $\beta_0$ is a single number rather than a vector, yet $\beta_e$, $\beta_m^T$ are the same $\beta_e$, $\beta_m^T$ [25]. We estimate $\beta_0$ using Equation (2) following Haneuse's et al. [38] optimization approach by using an estimated overall prevalence of AML for the years 1990–2007 and the simulated benzene exposure data. As input to this projection, we use two levels of benzene exposure: the minimum and maximum values of the benzene distribution. Having projected the binary AML variable, we inserted it into the BINaml column of the factual and counterfactuals table, along with the corresponding benzene exposures and other predictor values. Finally, we generated the factual and counterfactuals table by converting the benzene exposures from numerical to categorical (i.e., from a maximum value to 1 and from a minimum value to 0). We carry out this last conversion following the approach by Lebel [39] and the California of Environmental Health Hazzard Assessment's reference exposure level for benzene as 0.940 ppm (0.903 ppb) [40].

### 2.3. Distribution of Benzene Exposure

Heck et al. [23] provide a description of the distribution of outdoor benzene exposure, reproduced in Table 2.

**Table 2.** Outdoor benzene distribution [23].

| Agent | Mean/Standard Deviation | Inter-Quartile Range (IQR) | Minimum | Percentile | | | | Maximum |
|---|---|---|---|---|---|---|---|---|
| | | | | 10th | 25th | 75th | 90th | |
| Benzene, ppbv * | 1.268/0.830 | 1.197 | 0.151 | 0.410 | 0.591 | 1.788 | 2.574 | 4.600 |

* ppbv is parts per billion by volume.

For our benzene exposure data simulation, we use a normal distribution centered at 1.268 with a standard deviation of 0.830, a truncated minimum of 0.151, and a truncated maximum of 4.600. Heck et al. [25] calculate the risk of AML associated with one interquar-

tile range increase in benzene exposure; we normalize the benzene distribution parameters (mean, standard deviation, minimum, and maximum) by the IQR = 1.197 as well.

As for the observed prevalence of pediatric AML in California, we use an incidence of 46/19,255 (observed ratio of cases over controls) [25], reflecting their 1:20 matching.

In addition to the outdoor distribution, we also provide a causation analysis using an indoor benzene exposure distribution from locations near garages (Table 3) [41].

**Table 3.** Indoor benzene distribution [38].

| Agent | Mean/Standard Deviation | Inter-Quartile Range (IQR) * | Minimum | Percentile | | | | Maximum |
|---|---|---|---|---|---|---|---|---|
| | | | | 10th | 25th | 75th | 90th | |
| Benzene, ppbv | 5.650/2.825 | 1.197 | 0.700 | - | - | - | - | 12.000 |

\* Assumed the same as the outdoor distribution.

We normalized the parameters by the IQR of the outdoor benzene exposure to be able to compare the results using two different distributions.

### 2.4. Causation between Exposure to Benzene and Risk of Developing AML

We estimated the causal effects of the benzene exposure to AML using the average treatment effect and typical validation indicators obtained from Uplift modeling.

The average treatment effect (*ATE*) is a special case of an average partial effect for a binary explanatory variable. It is used to compare the effects in our randomized computational experiments. The *ATE* measures the difference in mean outcomes between units assigned to the study and units assigned to the control and is given by,

$$ATE = \frac{1}{N}\Sigma_i\left(y_{1(i)} - y_{0(i)}\right) \tag{3}$$

where the treatment effect for individual *i* is given by $y_{1(i)} - y_{0(i)} = \beta(i)$; the summation occurs over all N individuals in the population.

The value of the potential outcome *y(i)* must not be affected by how the treatment and exposure are assigned among all other individuals, according to the stable unit treatment value assumption that is required for the estimation of the ATE. Extrapolation based on strata must be assumed, or monotonicity instead, which denotes the absence of definers in the population. Therefore, if the experiment experiences non-compliance, the ATE can no longer be recovered. Instead, a local ATE can be obtained as the average treatment effect for a particular subpopulation, limiting its extrapolation. Generalization for the causal inference can be, hence, affected. The extrapolation based on ignobility or monotonicity, are difficult assumptions to verify; once the data is available, the foundations of the design of the experiment might reveal signals of homogeneity across groups that can verify them or not, for which further data analysis is required. A stronger argument for Uplift modeling can be made, as they provide a solution for isolating effects. Thus, Uplift models the difference between conditional response probabilities in both the treatment and control groups, clearly identifying groups of individuals on which an action or intervention will have the most 'positive effect'. A binary outcome is assumed for Uplift modeling, aligned to the odds ratio, as per the nature of the problem described in this work.

There are various Uplift modeling approaches. The response probabilities that differ between the study and control groups are used by the Two-Model approach to model the uplift. This leads to a methodology based on two models as these probabilities are calculated separately for each group. In Lo's approach (Figure 2), the independent variables change in logistic regression. The model is based (and learned) on one model; however, the predicted probabilities are calculated for both groups. For the calculation of the predicted probabilities, there is a dummy treatment variable in the test dataset, which is set to 0 for the control group and 1 for the treatment group [42].

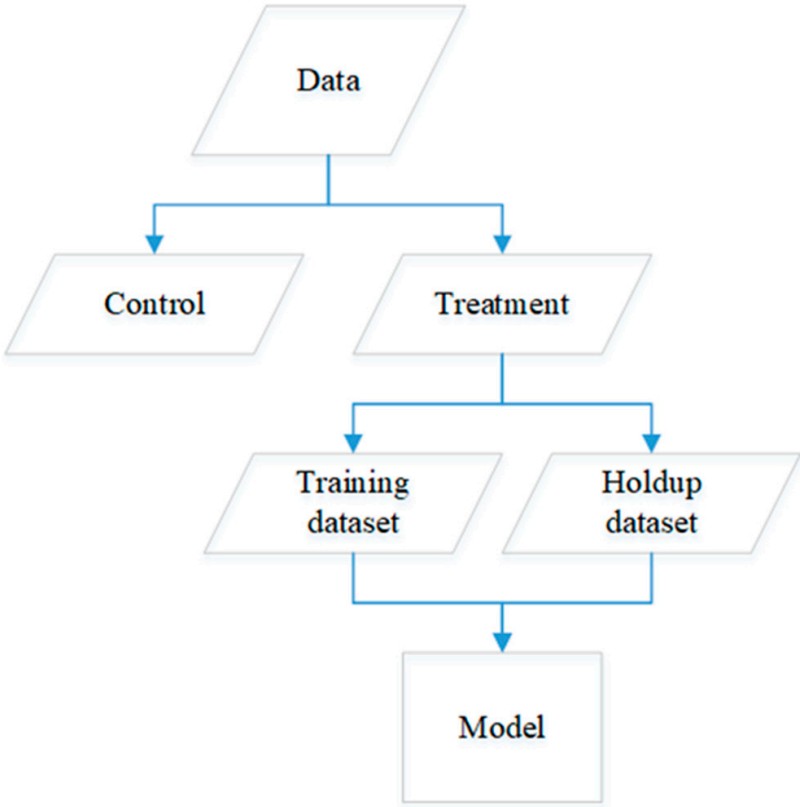

**Figure 2.** Uplift Two-model Approach.

The Uplift model validation is performed by selecting an appropriate cost function measuring the difference between the actual and predicted values of the response variable. In economics, the Gini coefficient is used to measure the model's goodness-of-fit. It is typically plotted to show the Lorenz curve where the predicted scores of the targeted observations are sorted in decreasing order. The extension of this curve and the Gini coefficient for Uplift modeling is called the Qini curve [43]. Figure 3 shows a typical Qini curve for an application in Econometrics.

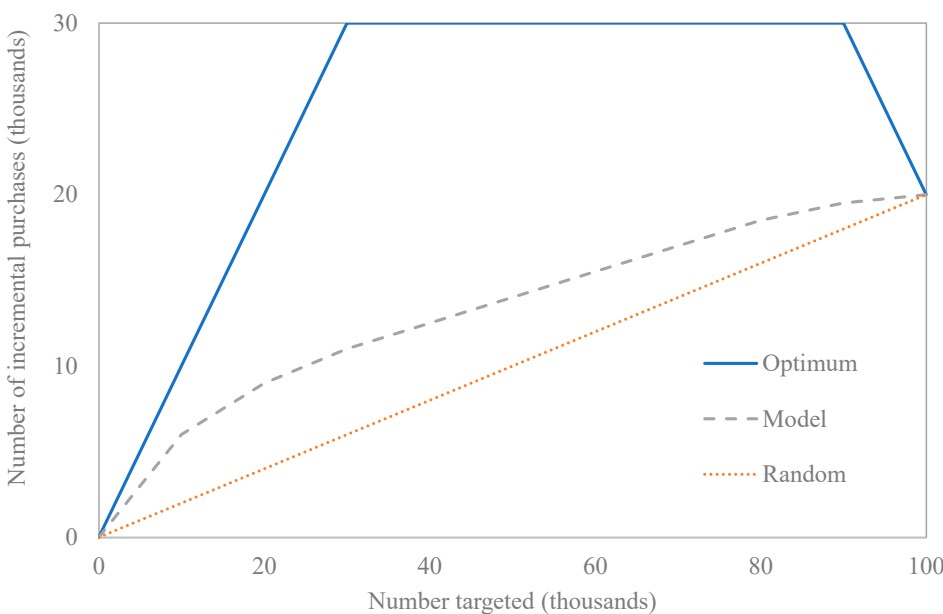

**Figure 3.** Typical Qini curve for Uplift model evaluation, used in Econometrics.

The *y*-axis shows the cumulative incremental gains, and the *x*-axis the proportion of the population targeted. There is an Uplift curve and a random curve based on the calculation of every segment. The Qini coefficient is the difference between the area under these curves. A positive Qini-coefficient represents a good performance of the model, while a value approximating zero represents a poorer performance.

### 2.5. Causation Codes

The Uplift data generation code (datagen.rmd) uses the βs provided by Heck et al. [23] and benzene distributions indoors and outdoors to find the corrected intercept $\beta\_0$ via optimization. This code, developed with the software R, is a modification of the program presented by Haneuse et al. [38]. The main results obtained include the ATE and the data table prepared as input for Uplift modeling.

We also developed an R code (*uplift.R*) for the Uplift modeling, using the library *tools4uplift*. The data is split into train and validation data for further fitting using the baseline model two-model estimator through the embedded function *DualUplift*. The results are interpreted from the QiniArea from the function *PerformanceUplift*.

Both codes datagen.r and uplift.rmd are available in Available online: https://github.com/CHE408UofT/AML_Uplift (accessed on 18 March 2023) [44].

### 2.6. Results Interpretation

ATE values equal to zero reveal no causation, while values different than zero reveal causation between exposure to benzene and the risk of developing childhood AML. We look at two different scenarios, as we included indoor and outdoor benzene distributions, looking at finding causation indoors and/or outdoors; therefore, we reported ATE values for both scenarios.

Regarding the Uplift modeling, the goodness-of-fit of the model is evaluated using the *QiniArea* function in R, which computes the area under the Qini curve. A positive value attests to a good performance of the model, while a value near 0 shows a worse performance.

## 3. Results

This section presents the results of using our ATE-Uplift framework to estimate the causation of benzene exposure (as an air pollutant) to childhood AML, through the analysis of the counterfactual nature of the OR-based relationships found indoors and outdoors.

### 3.1. Indoor Benzene Distribution

The results of generating a set of factual/counterfactuals tables for indoor benzene distribution to estimate their ATEs, show an average ATE of 0.203. This positive value reveals the causation between indoor exposure to benzene and the risk of developing AML in early childhood.

When analyzing the Uplift modeling results, we look at a variant of the Qini curve, representing the incremental uplift as a function of the proportion of the population target. Incremental Uplift measures whether an event would not have occurred without a specific interaction; hence positive Qinis attest to the goodness-of-fit of the Uplift model. Figure 4 shows a typical Qini curve for the indoor benzene distribution scenario.

The data is first partitioned into subsets that keep the same distribution of treated versus non-treated and responders versus non-responders values. The training was achieved using the formula *DualUplift* which fits the data using the two-model estimator or approach (logistic regression model), with splits of the data in 70% for the training and 30% for the validation. The first element of the *DualUplift* class is the baseline model fitted for nontreated individuals, and the second is the baseline model fitted for treated individuals. Using the two-model estimator, a baseline model is fitted for comparison purposes. Using the validation set, the function *predict* infers the uplift. Finally, to evaluate the quality of the baseline model, we plot the Qini curve, as shown in Figure 4. The Qini coefficients are single indexes of the Uplift model. The x-axis represents the fraction of targeted individuals

and the y-axis represents the incremental number of positive responses relative to the total number of targeted individuals. The straight line between the origin and (100, y-max) in Figure 4 represents a benchmark to compare the model's performance to a strategy that randomly targets subjects, as we explained before. In our case, the Qini coefficient is positive (with a value of 0.14) and outperforms random targeting. The uplift percentages are detailed in Figure 5. This reinforces the goodness-of-fit of the Uplift modeling approach, as the observed uplifts are ordered from highest to lowest.

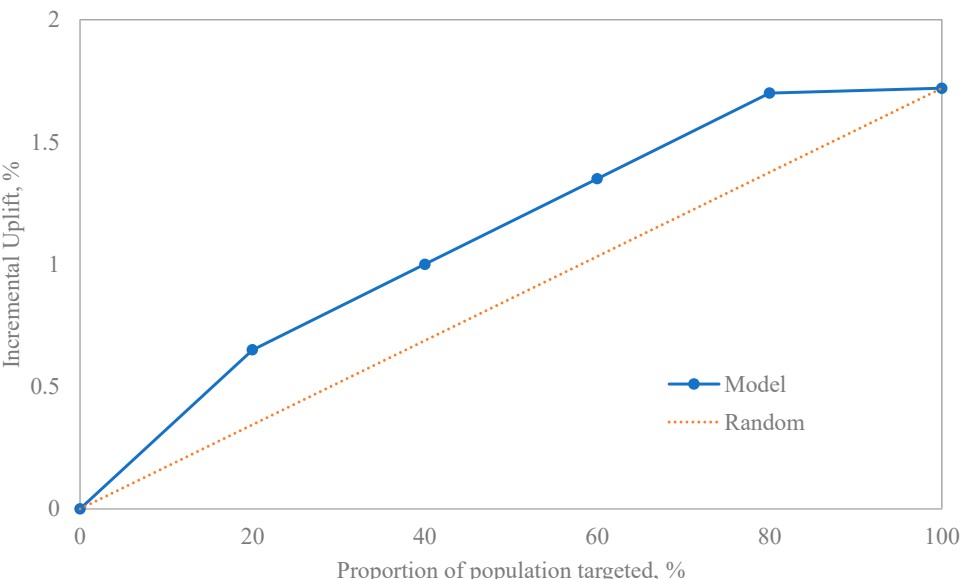

**Figure 4.** Qini curve for Uplift modeling for the validation data—indoor benzene distribution.

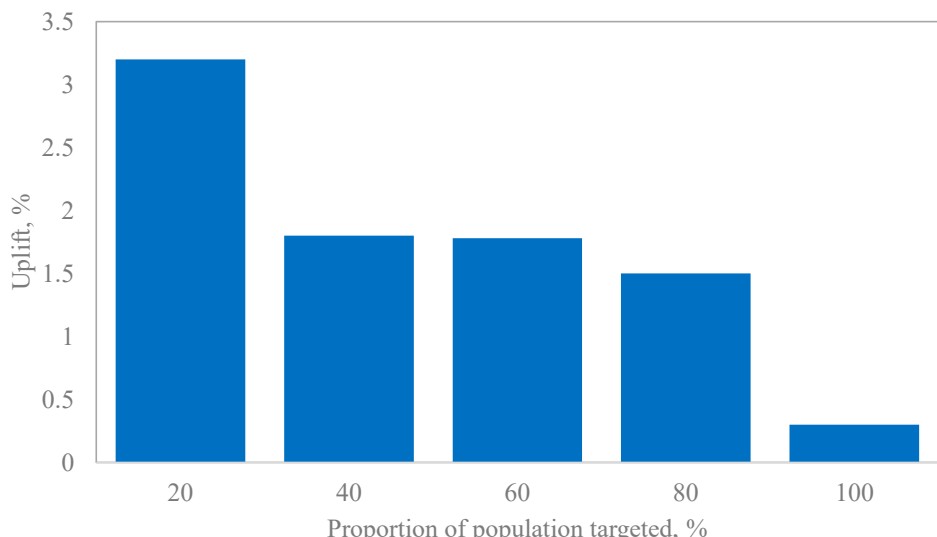

**Figure 5.** Uplift versus proportion of population targeted.

Our combined ATE/Uplift modeling framework allows us to conclude that there is a causation of benzene exposure (as an air pollutant) to AML in early childhood, analyzing the counterfactual nature of the OR-based relationship found indoors for the simulated data from Heck et al. [25] and the benzene exposure distribution considered [25].

### 3.2. Outdoor Benzene Distribution

The results of generating a set of factual/counterfactuals tables for outdoor benzene distribution to estimate their ATEs show an average value of zero, which indicates

no causation between outdoor exposure to benzene and the risk of developing AML in early childhood. The incremental uplift, in this case, is also zero at any proportion of the targeted population.

## 4. Discussion

The ATE-Uplift framework effectively predicts causation in health outcomes when fitted using ULR. Uplift modeling acts as a cross-validation technique of the ATE estimation, integrating its causal inference and machine learning features. Moreover, Uplift modeling ensures the goodness-of-fit of the regression. Thus, categorical factual/counterfactuals tables generated from the original data are input to obtain ATEs, with positive values revealing causation. Then, Uplift modeling is employed, where positive Qini coefficients confirm causation.

For the data considered by Heck et al. [25], their corresponding ULR fitting and distribution of benzene exposure [38], our framework was useful in revealing the causation between indoor exposure to benzene and the risk of developing childhood AML. In contrast, no causation was revealed when considering outdoor exposure. Nevertheless, it was observed that, for this scenario, low ATE values (<0.05) near zero are somehow related to inconsistent Uplift modeling results, as the corresponding Qini coefficients also show low values (<0.03), different than zero. In these instances, we recommend, using a third cross-validation method when reporting low ATE values.

While our preference is to have the original data to perform exploratory data analysis (EDA), evaluate the interaction between predictors, and consequently preselect -and later compare- applicable ML methods for estimating causation, in this work, we cautiously tested ATE-Uplift for generated data, with one predictor, benzene, as ATE and Uplift modeling are complementary methods for binary outcomes, working with factual and counterfactuals to consequently estimating causality. ML methods, although promising, must be carefully supported by a detailed EDA and a sample size evaluation. We noticed, for instance, that inconsistent Uplift modeling results are obtained when decreasing the sample size, an effect that must be minimized as per most ML predictors. Our future work will add to our framework an EDA that will allow us to observe trends among and within data groups, potentially fit and compare different surrogate models to establish the 'best' relationship between variables, and ultimately, perform a meta-analysis and comparison between causation-based ML methods. Some techniques to be explored in future works include meta-analysis and machine learning tools such as meta-learning causal structures and causal Bayesian networks.

It is important to note that the reliability of the results obtained through the application of our ATE-Uplift framework depends largely on the completeness of the data used and how well-defined the assumptions are. Thus, once the numerical and categorical data is available, it is recommended to perform data analysis to verify or not the main assumptions regarding the absence of definers in the population required by the ATE model and, therefore, frame the limitations of the model extrapolation.

Finally, we have assumed that there is no interaction between the benzene exposure and the rest of the predictors. Such interaction will be considered in future works adapting our framework once the data is available. ORs might be readjusted, and correlations between pollutants might be revisited for more accurate correlations representing the interaction within the strata predictors and benzene exposure.

Our research's scientific and practical novelty lies simply and effectively in estimating the causation of health outcomes using two levels of confirmation or cross-validation. We use ATE values and Uplift modeling, integrating causal inference and machine learning features. Causation is, per se, the main goal when evaluating health outcomes since confirmed relationships between variables might not indicate that the outcome is indeed the result of the occurrence of the other event(s). A reliable causation approach may lead to effective disease prevention, monitoring, and treatment.



## 5. Conclusions

In this work, we have presented the ATE-Uplift framework to predict causation in health outcomes. Uplift modeling and estimating ATE values effectively integrate causal inference and machine learning capabilities. We tested our framework to estimate the causation between benzene exposure and AML, verified when considering indoor exposure to this air pollutant. Causation is confirmed with two indicators, an ATE value different than zero and positive Qini coefficients. Further considerations to validate and universalize the use of this approach include an exhaustive exploratory data analysis (EDA) to observe trends that might allow confirming the assumptions for its applicability and analyzing the interaction between predictors, as its comparison with emerging ML methods being evaluated for causation.

**Author Contributions:** Conceptualization and visualization: D.G., R.T.-F., V.C.-A. and C.H.A. Data collection and methodology: D.G., R.T.-F. and A.N.-Z. Software: D.G. and R.T.-F. Formal analysis: D.G., R.T.-F. and V.C.-A. Validation: A.N.-Z., M.J. (Melanie Jeffrey), M.J. (Maria Jacome), J.B. and C.H.A. Computing resources: D.G., R.T.-F. and C.H.A. Writing—original draft: D.G. and R.T.-F. Preparation: D.G., R.T.-F. and A.N.-Z. Writing—review and editing: D.G., R.T.-F., V.C.-A., M.J. (Melanie Jeffrey), M.J. (Maria Jacome), J.B. and C.H.A. Supervision: D.G., R.T.-F., V.C.-A. and C.H.A.; funding acquisition: C.H.A. All authors have read and agreed to the published version of the manuscript.

**Funding:** This research was funded by a Healthy Cities Implementation Science Team Grants (LOI) 202110LT5 from the Canadian Institutes of Health Research.

**Data Availability Statement:** Not applicable.

**Acknowledgments:** We are grateful to Julia E. Heck from the UCLA Fielding School of Public Health, Department of Epidemiology, for the information and support provided throughout this research.

**Conflicts of Interest:** The authors declare that they have no known competing financial interests or personal relationships that could have appeared to influence the work reported in this paper.

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
