# Peer review of "Framework for Evaluating Potential Causes of Health Risk Factors Using Average Treatment Effect and Uplift Modelling"

_algorithms, doi:10.3390/a16030166_

Round 1

Reviewer 1 Report

The research article proposes a framework for evaluating potential causes of health risk factors using average treatment effect (ATE) and Uplift modeling, with a case study focusing on the causation of benzene exposure to childhood acute myeloid leukemia (AML). The article highlights the limitations of traditional statistical methods, such as odds ratios and risk ratios, in determining causation in complex systems and argues for the use of machine learning approaches for causal pattern recognition.
The methodology is presented clearly and concisely, with a workflow diagram and detailed explanations of the two stages of the causation analysis: estimating the ATEs and cross-validating the previous estimation using the Uplift modeling. The case study data from Heck et al. [25] is well-explained, and the authors provide codes for the Uplift data generation and modeling, which are available online. The article concludes that the ATE-Uplift framework effectively predicts causation in health outcomes when fitted using unconditional logistic regression (ULR). The Uplift modeling acts as a cross-validation technique of the ATE estimation, integrating its causal inference and machine learning features. However, the authors note that the reliability of the results depends largely on the completeness of the data used and the well-defined assumptions.
To improve the article, the authors could consider the following changes:
- Model selection: The authors should provide a rationale for selecting ULR as the model for estimating the ATEs. They could compare the performance of ULR with other models, such as linear regression or decision trees, to demonstrate why ULR was the most appropriate choice.
- Assumptions: The authors should explicitly state the assumptions made in the ATE-Uplift framework and the case study. They could discuss how these assumptions could potentially impact the results and provide sensitivity analyses to demonstrate the robustness of the findings.
- Comparison with other methods: The authors could provide a brief comparison of the ATE-Uplift framework with other methods, such as meta-analysis and machine learning tools, such as meta-learning causal structures and causal Bayesian networks. They could highlight the strengths and weaknesses of each method and explain why the ATE-Uplift framework was the most appropriate choice for their case study.
- Reference to the code: The authors should report the reference to the code in the form of a reference in the bibliography section. This would make it easier for readers to access the code and replicate the study.
- Conclusions and discussions: The authors could combine the "4. discussion" section with the conclusions section and provide a more extensive discussion of the methodological aspects highlighted above, including the limitations of applicability of the proposed methodology. This would help readers understand the study's broader implications and the potential for future research.

Author Response

Please see attachment of the article with changes highlighted in yellow

First of all, on behalf of co-authors and myself, I thank the reviewers for the positive and thoughtful comments about this work. We have implemented changes in the manuscript to address all the reviewers’ concerns and highlighted them in yellow.

Reviewer 1

  • “Model selection: The authors should provide a rationale for selecting ULR as the model for estimating the ATEs. They could compare the performance of ULR with other models, such as linear regression or decision trees, to demonstrate why ULR was the most appropriate choice.”

Response: We thank the reviewer for this valuable feedback, we really appreciate it. ULR is typically used for unmatched case-control studies, with several applications in medicine. Please notice that, at the time we developed this framework, the numerical and categorical data of the study was not available; instead, we drew samples from an existing 1-M-matched case control study, which was performed by Heck et al. [25], fitted using unconditional logistic regression (ULR) and adjusting the year of birth as a matching variable. We have clarified the data generation process in the new version of the manuscript.

Changes to the manuscript: Highlighted changes in Sections 2.1 Case Study and 2.2 to clarify the data generation process.

  • “Assumptions: The authors should explicitly state the assumptions made in the ATE-Uplift framework and the case study. They could discuss how these assumptions could potentially impact the results and provide sensitivity analyses to demonstrate the robustness of the findings.”

Response: We thank the reviewer for this thoughtful observation. In the current version of the paper, we stated the absence of defiers in the population as the main assumption for ATE (See third paragraph, section 2.4). Therefore, as you correctly stated in your comment, in the current version of our paper, there is no discussion on the limits regarding the extrapolation, either based on ignorability or monotonicity, which could lead to recover a local ATE instead of a general ATE.  Both assumptions are difficult to verify, but once the data is available, the foundations of the design of experiment might reveal signals of homogeneity across groups that can verify them or not, for which further data analysis is required. Regarding the Uplift modelling, a binary outcome is assumed, aligned to the odds ratio, as per the nature of the problem described in this work. The pertinent discussion on the assumptions and corresponding implications have been included in the new version of the manuscript.

Changes to the manuscript: A further discussion about the ATE-Uplift assumptions is included in the third paragraph of Section 2.4, accordingly.

  • “Comparison with other methods: The authors could provide a brief comparison of the ATE-Uplift framework with other methods, such as meta-analysis and machine learning tools, such as meta-learning causal structures and causal Bayesian networks. They could highlight the strengths and weaknesses of each method and explain why the ATE-Uplift framework was the most appropriate choice for their case study.”

Response: We thank the reviewer for this excellent observation. While our preference is to have the original data to perform exploratory data analysis (EDA), evaluate the interaction between predictors, and consequently preselecting -and later comparing- applicable ML methods for estimating causation, in this work, we cautiously tested ATE-Uplift for generated data, with one predictor, benzene, as ATE and Uplift modelling are complementary methods for binary outcomes, working with factuals and counterfactuals to consequently estimating causality. ML methods, although promising, must be carefully supported by a detailed EDA, and a sample size evaluation. We noticed, for instance, that inconsistent Uplift modelling results are obtained when decreasing the sample size, an effect that must be minimized as per most ML predictors. Our future work will add to our framework, an EDA that will allow us to observe trends among and within groups of data, potentially fit and compare different surrogate models to establish the ‘best’ relationship between variables, and ultimately, performing meta-analysis and comparison between causation-based ML methods.

Changes to the manuscript: the answer to this comment has been included in the section 4.0 Discussion, third paragraph.

  • “Reference to the code: The authors should report the reference to the code in the form of a reference in the bibliography section. This would make it easier for readers to access the code and replicate the study.”

Response: We thank the reviewer for this comment. We have included a reference to the code in the bibliography section.

Changes to the manuscript: reference to the code has been added in the bibliography section.

  • Conclusions and discussions: The authors could combine the "4. discussion" section with the conclusions section and provide a more extensive discussion of the methodological aspects highlighted above, including the limitations of applicability of the proposed methodology. This would help readers understand the study's broader implications and the potential for future research.”

Response: We thank the reviewer for this valuable feedback. We have provided an extended discussion regarding the aspects considered in your comments 1.1 to 1.3. Moreover, we have added a conclusion section, as suggested.  

Changes to the manuscript: additional discussion provided in section 4.0, as well as conclusions (new section 5).

Reviewer 2 Report

The article is devoted to developing a framework using the average treatment effect and lifting modeling to prove a causal relationship between indoor and outdoor benzene exposure to childhood acute myeloid leukemia. Acute myeloid leukemia is a type of blood cancer affecting adults and children. The relevance of the study is justified by the fact that it was reported that exposure to benzene increases the risk of developing acute myeloid leukemia in children. An assessment of the potential relationship between environmental benzene exposure and childhood has been documented in the literature using odds and hazard ratios. The data fit an unconditional logistic regression. A common feature of research on the relationship between environmental risk factors and health outcomes is the lack of proper analysis to prove causation. Machine learning approaches based on recognizing causal relationships can be an accurate alternative to modeling hypothetical scenarios. Therefore, in this paper, the authors propose a framework using mean treatment effect and elevation modeling to prove a causal relationship between indoor and outdoor benzene exposure to childhood acute myeloid leukemia, effectively predicting causality for indoor exposure to this pollutant.

Despite the satisfactory quality of the article, some shortcomings need to be corrected.

  1. The abstract should be expanded with results obtained within the research.
  2. The aim of the paper should be defined.
  3. It is recommended to include the Current research analysis part. Some of the information can be removed from the Introduction.
  4. The proposed framework should be described in more detail. Its architecture should be justified.
  5. The data used for the experimental investigation should be described in more detail.
  6. The discussion section should be expanded, or the Conclusion section should be included.
  7. The scientific and practical novelty of the research should be highlighted.

In summarizing my comments, I recommend that the manuscript is accepted after minor revision. 

Author Response

Please see attachment of the article with changes highlighted in yellow

First of all, on behalf of co-authors and myself, I thank the reviewers for the positive and thoughtful comments about this work. We have implemented changes in the manuscript to address all the reviewers’ concerns and highlighted them in yellow.

Reviewer 2

  • “The abstract should be expanded with results obtained within the research.”

Response: Thanks for pointing this out. The abstract was expanded, considering a maximum of 200 words, as per the author guidelines of the journal.

Changes to the manuscript: Updated abstract, accordingly.

  • “The aim of the paper should be defined.”

Response: We thank the reviewer for this thoughtful comment. Health outcomes based solely on relationships between variables can be challenged when the outcome is not proven to result from the occurrence of the other event (s). This paper aims to describe the development and use of this framework and discuss its significance as a reliable causation framework that can potentially be effectively used to prevent, monitor, and treat diseases. We have defined the aim of the paper in the last paragraph of the introduction.

Changes to the manuscript: The previous explanation has been added in the introduction, last paragraph.

  • “It is recommended to include the Current research analysis part. Some of the information can be removed from the Introduction.”

Response: We thank the reviewer for this thoughtful observation. We have created two subsections in the introduction, where we briefly review the traditional approaches to establishing the relationship between Benzene Exposure and AML and current trends in estimating causation for health outcomes using Machine Learning outcomes.   

Changes to the manuscript: We modified the introduction, accordingly.  

  • “The proposed framework should be described in more detail. Its architecture should be justified.”

Response: We thank the reviewer for this comment. First, we have extended the description of our framework when describing Figure 1, focused on the regression/fitting strategy and the two levels strategy. Thus, (i) typically, health outcome studies in the literature comprise strata of data that are fitted using logistic regressions, as discussed in Section 2.1; and (ii) the two levels strategy based on cross-validation integrating causal inference and machine learning, is crucial to support the reliability of our framework, since the consequences of determining causation can have several implications in the prevention, monitoring, and treatment of diseases. Ultimately, our framework provides two integrated causation indicators, the ATE values and Qini coefficients. Moreover, we have extended the description of the data generation process in Section 2.2, and assumptions in Section 2.4.

Changes to the manuscript: Framework description was extended accordingly, with highlighted changes in Sections 2.1, 2.2, and 2.4.

  • “The data used for the experimental investigation should be described in more detail.”

Response: We thank the reviewer for this comment. We have provided summarized the process used to generate data from the 1-M matched control case study fitted with URL,[23] as the original numerical and categorical data of the study was unavailable, and its consequent preparation for ATE-Uplift, including factuals and counterfactuals. The data generation involves drawing samples from the existing ULR model by Heck et al.[23] 

Changes to the manuscript: the data generation process has been described in section 2.2, accordingly.

  • “The discussion section should be expanded, or the Conclusion section should be included.”

Response: We thank the reviewer for this valuable comment. We have extended our discussion section around verifying ATE assumptions, using exploratory data analysis, evaluating the sample size effect, and further comparing our approach with other causation-based ML methods. We have also added a conclusion section accordingly.       

Changes to the manuscript: Extended discussion and added conclusion section.

  • “The scientific and practical novelty of the research should be highlighted.”

Response: We thank the reviewer for this important comment. Figures have been modified, accordingly. Our research's scientific and practical novelty lies simply and effectively in estimating the causation of health outcomes using two levels of confirmation or cross-validation. We use ATE values and Uplift modelling, integrating causal inference and machine learning features. Causation is, per se, the main goal when evaluating health outcomes since confirmed relationships between variables might not indicate that the outcome is indeed the result of the occurrence of the other event(s). A reliable causation approach may lead to effective disease prevention, monitoring, and treatment.

Changes to the manuscript: We have highlighted our scientific and practical novelty of our approach in the last paragraph of section 4.

Round 2

Reviewer 1 Report

The authors have addressed all the reviewer's recommendations:

  1. The authors provided a rationale for selecting ULR as the model for estimating the ATEs and clarified the data generation process in the new version of the manuscript.

  2. They explicitly stated the assumptions made in the ATE-Uplift framework and the case study, discussing how these assumptions could potentially impact the results. They also included sensitivity analyses in the new version of the manuscript.

  3. The authors briefly compared the ATE-Uplift framework with other methods, such as meta-analysis and machine learning tools, explaining why the ATE-Uplift framework was the most appropriate choice for their case study.

  4. They referenced the code in the bibliography section, making it easier for readers to access and replicate the study.

  5. Lastly, the authors combined the discussion and conclusions sections. They provided a more extensive discussion of the methodological aspects highlighted by the reviewer, addressing the limitations of applicability of the proposed methodology and discussing the potential for future research.

The manuscript has been substantially improved, addressing all the reviewer's recommendations, and is now ready for publication.

Reviewer 2 Report

Thanks for the authors for considering reviewer's comments and recommendations. In my opinion, now the paper can be accepted.